# Assessing the viability of transplanted gut microbiota by sequential tagging with D-amino acid-based metabolic probes

Wei Wang[1,2], Liyuan Lin[1,3], Yahui Du[3], Yanling Song[1], Xiaoman Peng[1], Xing Chen [2] & Chaoyong James Yang [1,3]

Currently, there are more than 200 fecal microbiota transplantation (FMT) clinical trials worldwide. However, our knowledge of this microbial therapy is still limited. Here we develop a strategy using sequential tagging with D-amino acid-based metabolic probes (STAMP) for assessing the viabilities of transplanted microbiotas. A fluorescent D-amino acid (FDAA) is first administered to donor mice to metabolically label the gut microbiotas in vivo. The labeled microbiotas are transplanted to recipient mice, which receive a second FDAA with a different color. The surviving transplants should incorporate both FDAAs and can be readily distinguished by presenting two colors simultaneously. Isolation of surviving bacteria and 16S rDNA sequencing identify several enriched genera, suggesting the importance of specific bacteria in FMT. In addition, using STAMP, we evaluate the effects on transplant survival of pre-treating recipients using different antibiotics. We propose STAMP as a versatile tool for deciphering the complex biology of FMT, and potentially improving its treatment efficacy.

[1] Institute of Molecular Medicine, Renji Hospital, Shanghai Jiao Tong University School of Medicine, Shanghai 200127, China. [2] College of Chemistry and Molecular Engineering, Peking-Tsinghua Center for Life Sciences, Synthetic and Functional Biomolecules Center, and Key Laboratory of Bioorganic Chemistry and Molecular Engineering of Ministry of Education, Peking University, Beijing 100871, China. [3] Collaborative Innovation Center of Chemistry for Energy Materials, The MOE Key Laboratory of Spectrochemical Analysis and Instrumentation, State Key Laboratory of Physical Chemistry of Solid Surfaces, Department of Chemical Biology, College of Chemistry and Chemical Engineering, Xiamen University, Xiamen 361005, China. Correspondence and requests for materials should be addressed to X.C. (email: xingchen@pku.edu.cn) or to C.J.Y. (email: cyyang@xmu.edu.cn)

The past decade has witnessed a great leap forward in our understanding of the diverse physiological and pathological functions of the gut microbiota[1,2]. The ever-increasing interest in microbiota research has been further motivated by the development of fecal microbiota transplantation (FMT) as a potential therapy for a variety of diseases[3], including *Clostridium difficile* infection[4], inflammatory bowel disease[5], irritable bowel syndrome[6], and some extra-intestinal disorders[7]. Presently, there are more than 200 FMT clinical trials completed or ongoing worldwide (www.clinicaltrials.gov/ct2/results/details?term=fecal +microbiota+transplantation), however, we still have little knowledge about how the transplanted bacteria survive, colonize, and function[8,9]. One reason for this poor knowledge is the absence of a feasible method to track the transplanted microbiotas and evaluate their viabilities. Although the subsistence and colonization of the transplanted microbiota can be investigated longitudinally by metagenomic sequencing and bioinformatic analyses, the procedures are complicated and expensive[10–12].

We envisioned that tracking the transplanted microbiotas by fluorescent imaging would be a promising strategy if one could fluorescently label the microbiota. Because most gut bacteria is not yet amenable to genetic manipulations, the use of foreign fluorescent proteins for tracking has only met with limited success[13,14]. Another method is the use of fluorescence in situ hybridization (FISH) probes to label bacterial rRNA. Unfortunately, FISH requires fixation of cells, and cannot be used in following living gut bacteria[15]. Recently, metabolic glycan labeling

with azidosugars followed by click reaction with an alkyne-functionalized fluorophore has been employed in visualizing living commensal gut bacteria without the use of genetic engineering[16]. However, this approach can only label a specific group of gut microbes[16,17]. Alternatively, analogs of D-amino acids (DAAs) functionalized with a fluorophore at the side chain (i.e., fluorescent D-amino acids, FDAAs) have been developed for metabolic labeling of bacterial peptidoglycans (PGNs)[18–20]. PGNs are ubiquitous among most bacteria, which use DAAs as essential building blocks. FDAAs have been shown to be well-tolerated by the enzymes involved in PGN construction[18], and can quickly label bacteria with high efficiency. Recently, FDAAs have been used for in vivo microbial labeling[19]. In addition, chronological incorporation of multiple FDAAs has been demonstrated in bacteria cultured in vitro[18,20].

Here, we report the development of a sequential tagging with DAA-based metabolic probes (STAMP) strategy for fluorescent tracking and assessing the viabilities of transplanted microbiotas (Fig. 1). STAMP exploits the fact that FDAAs are metabolically incorporated into PGNs only in living bacteria. The gut microbiotas are labeled with FDAA in vivo in the donor mice. After transplantation into the recipient mice, a second FDAA with a distinct color is administered. Only the surviving and viable bacteria among the transplants contain both FDAA labels, which can be readily detected by two-color fluorescence microscopy and flow cytometry. STAMP provides a method for visualizing transplanted microbiotas and evaluating their viabilities in the recipients, facilitating further

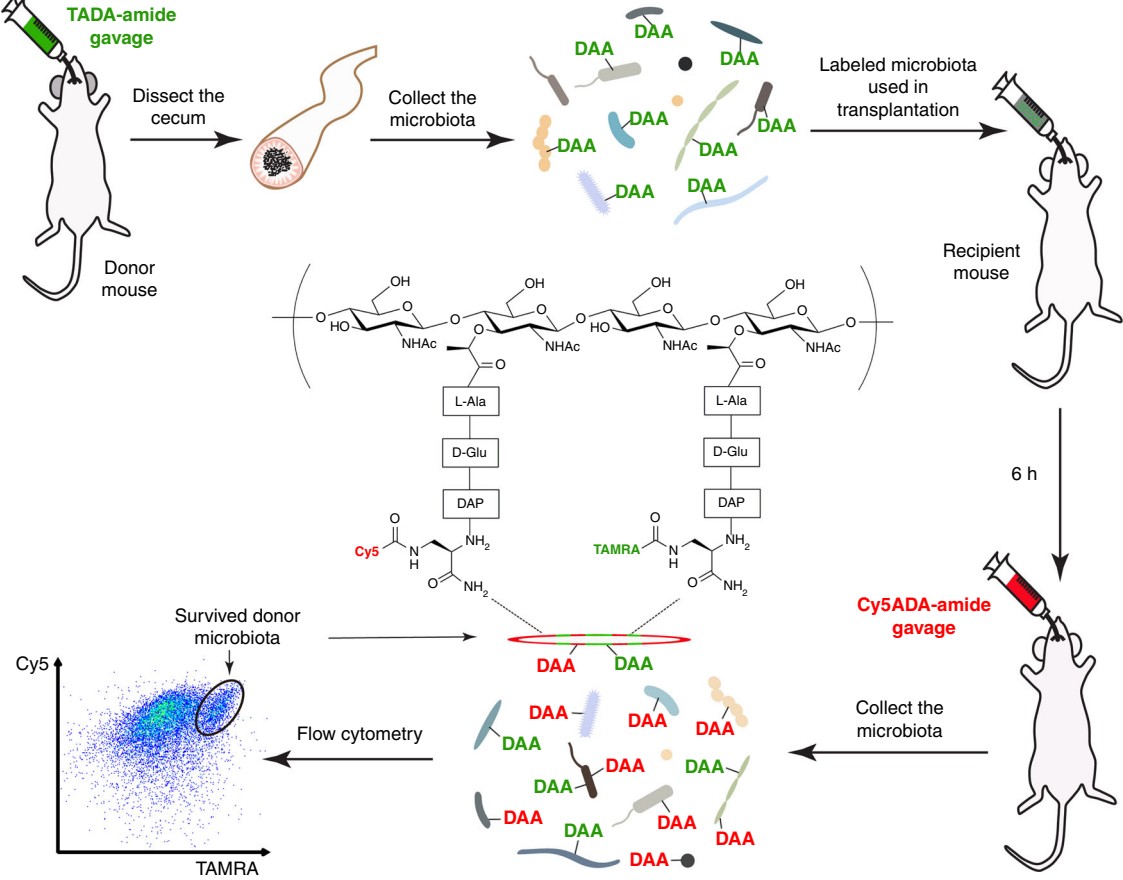

**Fig. 1** STAMP for assessing the viability of transplanted gut microbiotas. The donor mouse gut microbiota labeled in vivo by the first FDAA probe (TADA-amide) was intragastrically administered to the recipient mouse, which was then given a second FDAA probe (Cy5ADA-amide) by gavage 6 h after the transplantation. The recipient's gut microbiota was collected, and analyzed by two-color fluorescence microscopy and flow cytometry. The bacteria labeled by both probes were the transplants that survived in the recipient's gut, bacteria labeled with only TAMRA were probably those did not survive during transplantation, and bacteria labeled only by Cy5 were the recipient's original gut microbiota

investigations on FMT's functioning mechanisms and potentially improving its efficacy as a clinical treatment.

## Results

**In vivo labeling of the gut microbiotas in donor mice**. To label the bacteria with high coverage and intensity, we first optimized the gut microbiota in vivo labeling procedures. FDAAs with different protecting groups on the α-carboxyl group have previously been shown to label bacteria with varied efficiencies in vitro[21,22]. We therefore evaluated three TAMRA (tetramethylrhodamine)-bearing FDAAs, TAMRA-amino-D-alanine (TADA), TADA-amide, and TADA-ester, which possessed no protection, an amide moiety, and a methyl ester group on the carboxyl group, respectively (structures shown in Fig. 2a). Following two gavages to a group of specific-pathogen-free (SPF) C57BL/6 mice, their cecal microbes were collected and analyzed by fluorescence microscopy and flow cytometry. All three probes showed strong labeling of gut microbes (Fig. 2b and Supplementary Fig. 1), and TADA-amide exhibited the highest labeling coverage (Fig. 2c). This is consistent with the previous in vitro labeling results, where amide-protected FDAAs showed stronger labeling in most of the tested bacterial species compared to other probes[21]. This might be because the amide moiety of the incorporated FDAAs renders PGN resistance to degradation in some bacterial species[21]. As expected, many of the unlabeled bacteria were found to be dead bacteria (Supplementary Fig. 2), since metabolic activity was essential for FDAA incorporation. By using a FISH probe targeting *Bacteriodetes*, the major Gram-negative phylum in gut microbiota, we also found that some *Bacteriodetes* were not labeled by the FDAA (Supplementary Fig. 3).

For the purpose of sequential labeling, we synthesized Cyanine 5 (Cy5)-conjugated DAA with the amide protection (Cy5ADA-amide) as the second FDAA. It was reported that smaller fluorophores on FDAA side chains had better labeling efficiency in Gram-negative bacteria[23]. To determine whether the relatively big size of Cy5 would affect the labeling coverage of gut microbiota, we used two FDAA-amide probes, TADA-amide and Cy5ADA-amide, in the same gavage. The labeling signals from the two probes were highly overlapped with similar labeling coverages (Supplementary Fig. 4), indicating that Cy5ADA-amide had a similarly high labeling efficiency for gut microbiotas. Of note, this does not completely exclude the possibility of an improved labeling of this small subgroup of Gram-negative bacteria by using FDAA containing a smaller fluorophore[23]. Considering the low ratio of the unlabeled living bacteria and the fact that FDAAs with relatively small fluorophores are often green-emitting, where the gut bacteria have strong autofluorescence, we decided to use TADA-amide and Cy5ADA-amide in the following studies.

Using the TADA-amide probe, we optimized the gavage procedures for microbiota labeling, and found that two gavages with a 3 h interval showed the highest labeling signal (Fig. 2d), and the labeling was dose-dependent (Supplementary Fig. 5). Compared to a previous report where 3 mM of FDAAs was used in gavage[19], we used a much lower concentration of FDAA (1 mM, thus less disturbance to the microbiota) in the following experiments, which still achieved strong fluorescent labeling.

**Labeling of transplanted microbiotas in recipient mice**. With the fluorescently labeled gut microbiotas in hand, we used them in a microbiota transplantation mouse model for transplant-tracking. The TADA-amide-labeled cecal microbiotas were given to a group of recipient mice by gavage, where the transplanted bacteria could be clearly visualized on the tissue slices of both small and large intestines (Fig. 2e). This allows monitoring of the transplants by fluorescence imaging and should facilitate the FMT studies where the biogeography of transplants is of interest[24].

Taking advantage of this FMT model where the transplanted bacteria could be readily differentiated from the recipient's original microbiota, we next examined the viability of the transplants by evaluating the metabolic incorporation of a second FDAA with a distinct color. Six hours after the transplantation, the recipient mice received two Cy5ADA-amide gavages with an interval of 3 h. The 6 h waiting time was determined by the fluorescence decay of the donor microbiota (Supplementary Fig. 6). Therefore, 12 h in total after transplantation, the recipient's microbiota was collected and analyzed. Using confocal

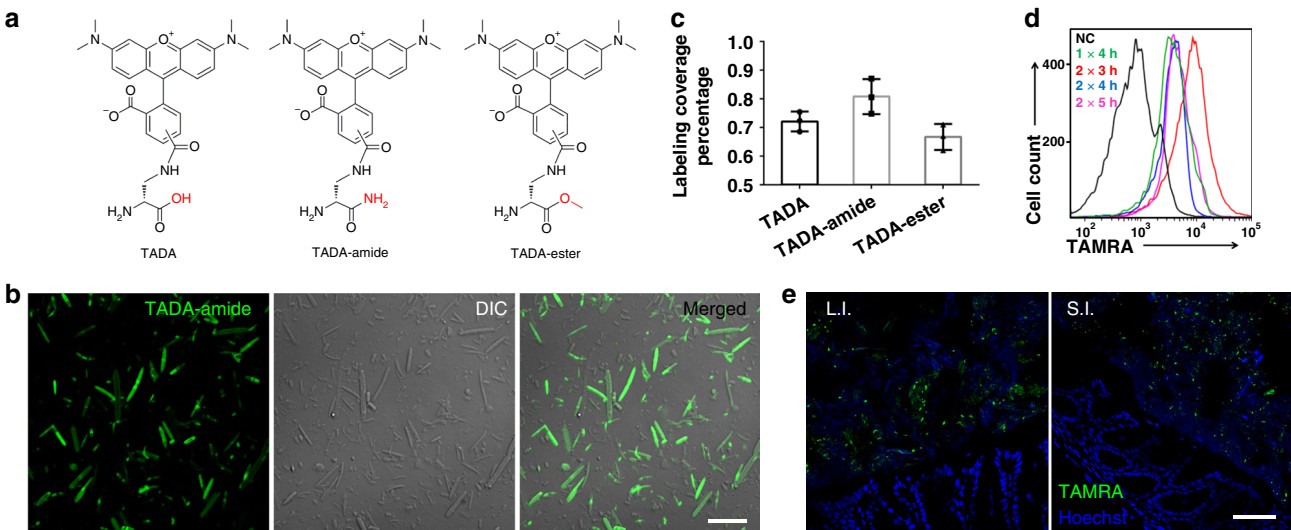

**Fig. 2** The FDAA probes used in this study and their labeling of gut microbiota in vivo. **a** Structures of TADA, TADA-amide, and TADA-ester. **b** Confocal fluorescence images of gut microbes from mice administered with TADA-amide. DIC, differential interference contrast. Scale bar, 10 μm. **c** Statistical analysis of the labeling coverage for gut microbes labeled with TADA, TADA-amide, and TADA-ester. Mean ± s.d. are presented for $n = 3$. **d** Flow cytometry analysis of TADA-amide-labeled gut microbiota with different labeling time. NC negative control; 1 × 4 h, one gavage and microbiotas collected 4 h later; 2 × 3 h, two gavages with a 3 h interval, and microbiotas collected 3 h after the second gavage. **e** Confocal fluorescence imaging of the transplanted gut microbiota (green) on the tissue sections of the recipient mouse's large intestine (L.I.) and small intestine (S.I.). Hoechst 33342 (blue) was used for nuclear counterstain. In **b**, **e**, representative results from three independent experiments are shown. Scale bar, 50 μm

microscopy, the distributions of the two colors on gut bacteria were clearly observed (Fig. 3a). The dually labeled bacteria accounted for ~8.6% of the gut microbiota from the recipient mice as shown by flow cytometry analysis (Fig. 3b). Their TADA-amide labeling indicated that they were from the donor microbiota, and the Cy5ADA-amide labeling demonstrated their metabolic activity in the recipient's gut. Taken together, these dually-labeled bacteria were most likely the transplanted bacteria that managed to survive in the recipient's gut.

We then collected the two-colored bacteria by fluorescence activated cell sorting (FACS) for sequencing (Supplementary

Fig. 7). The 16S rDNA analyses indicated that several bacterial genera were enriched in this population, including Gammaproteobacteria (*Acinetobacter* and *Escherichia/Shigella*), *Clostridium XIVb*, and *Butyricicoccus* (Fig. 3c and Supplementary Fig. 8), suggesting their high viabilities during transplantation. The survival of some Gammaproteobacteria in FMT is probably due to their resilience to acid stress[25], and it is very encouraging to find that *Clostridium XIVb* and *Butyricicoccus* were also better survivors, since they were generally considered to be beneficial to their hosts[26,27]. The fact that several bacterial genera survived better than others during transplantation implied that the

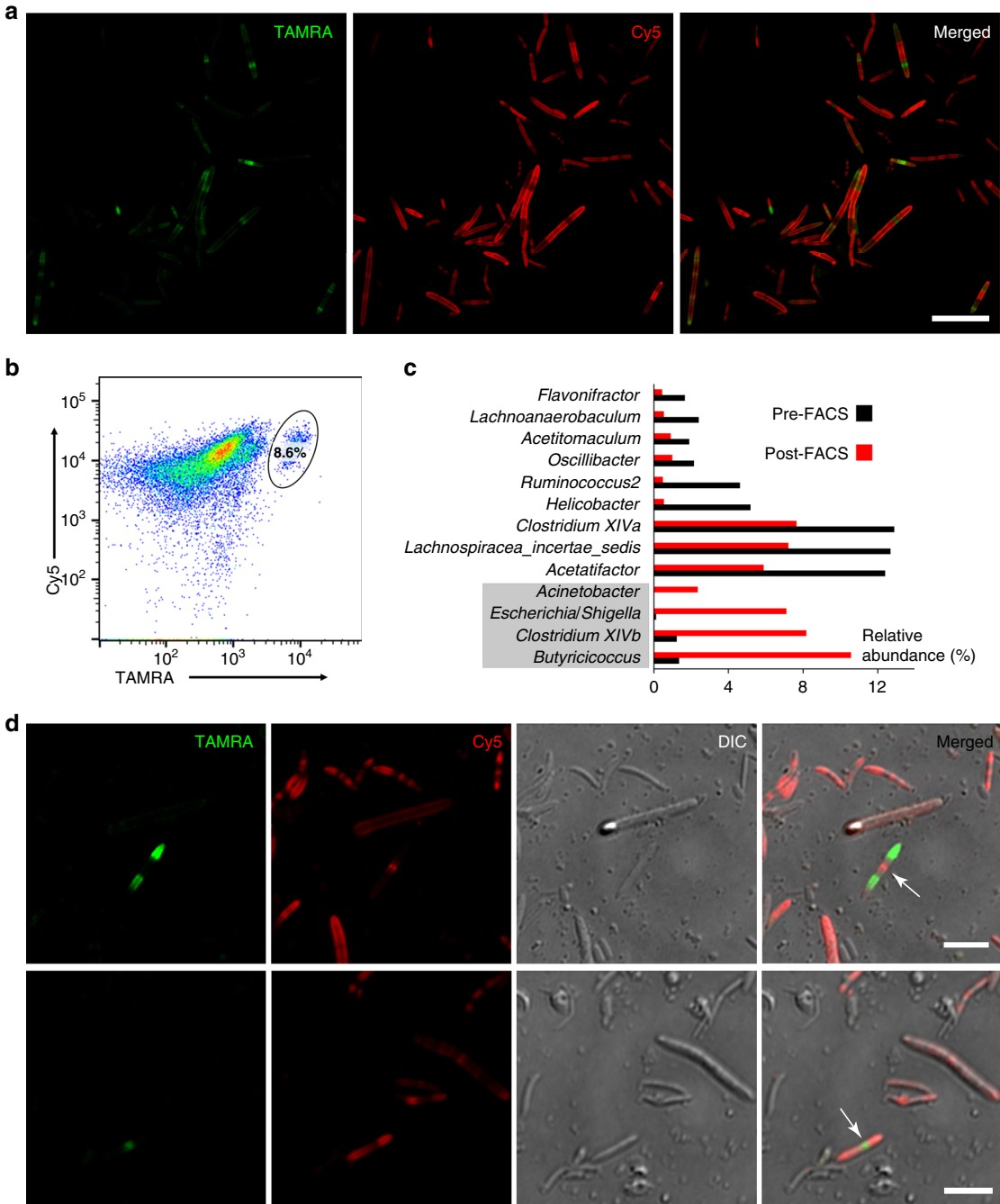

**Fig. 3** Two-color fluorescence analyses of the transplanted microbiotas. **a** Two-color fluorescence microscopy analyses of the transplanted microbiota from the recipient mice. Scale bar, 20 μm. **b** Flow cytometry analysis of the transplanted microbiotas from the recipient mice. The inserted ellipse indicates the survived transplants labeled with both colors, which accounts for 8.6% of the recipient mice's microbiota. **c** 16S rDNA sequencing of the bacteria before and after sorting uncovered that several bacterial genera were enriched in the transplantation survivors. **d** Gut bacteria with two classical dividing patterns were revealed by two-color fluorescence microscopy. Scale bars, 5 μm. Representative data from three independent experiments are shown

functioning of FMT might be mainly carried out by a specific subgroup of the transplanted microbiota. In addition, the abundance of these bacteria in the donor microbiotas may need to be considered when choosing donors for FMT patients.

Besides the survived transplants, most of the bacterial population was single-colored. Those showing only TADA-amide labeling were probably the transplanted bacteria that died during transplantation, since they had little metabolic activities (Cy5 signals) in the recipient's gut. The bacteria only labeled with Cy5ADA-amide were most likely the recipient's original gut microbiota (Supplementary Fig. 9).

Intriguingly, distinct reproduction patterns of gut bacteria were observed for the dually labeled bacteria. The bacterium with red labeling at the center and green at the poles (Fig. 3d, top panel) probably reproduced via binary fission with the newly synthesized cell walls in the middle. The other bacterium (Fig. 3d, bottom panel) very likely reproduced in a pattern where the newly synthesized cell wall lies at the poles. STAMP, therefore, provides a means of studying different division patterns in gut bacteria, especially for those that are unculturable in vitro.

### Evaluation of the survival of specific bacteria during transplantation.

In addition to assess the whole microbiota, STAMP could also be used for evaluating the survival of specific bacteria during transplantation (scheme shown in Fig. 4a). We chose *Escherichia* as an example, because it was enriched in the survived transplants (Fig. 3c). Two *Escherichia coli* strains were tested: K12, a commensal strain, and Nissle 1917, a probiotic strain that had been extensively studied[28]. Two strains were first labeled by TADA in vitro and then given to two groups of C57BL/6 mice by gavage, respectively. A second round of Cy5ADA labeling was performed 6 h after the transplantation. As expected, both strains could survive in the recipients' gut with the survival rates of ~5.3% and ~13.6% for Nissle 1917 and K12, respectively (Fig. 4b, c and Supplementary Fig. 10), indicating that different bacterial strains might have varied viabilities in the recipients' gut. The in vivo viabilities of other probiotic bacteria could also be examined using STAMP strategy.

### STAMP for tracking fecal microbiota from both mouse and human.

In clinics, fecal microbiotas are usually used in transplants. We therefore collected the fecal pellets of donor mice that were labeled with TADA-amide. Fluorescence microscopy showed strong labeling of the bacteria (Supplementary Fig. 11a). The collected fecal bacteria were used in FMT as donor microbiota. After a second FDAA (Cy5ADA-amide) labeling, a subgroup of gut bacteria were found to be dually labeled (Supplementary Fig. 11b), demonstrating the effectiveness of our STAMP method for tracking the viability of fecal bacteria in FMT. Furthermore, by using a previously reported human gut microbiota in vitro culture system[17], the bacteria in the human feces were strongly labeled by TADA-amide as well, demonstrating its feasibility with human microbiotas (Supplementary Fig. 12).

### Antibiotic preconditioning effects evaluated by STAMP.

In clinics, patients awaiting FMT often received an intense antibiotic treatment, which was meant to disrupt the existing gut flora and potentially improve FMT's efficacy[29]. However, the validity of this preconditioning has been controversial[30]. To evaluate its effectiveness, we pre-treated four groups of recipient mice with four different antibiotics for 10 days before transplantation, and compared the survival rates of the transplanted bacteria. Intriguingly, only polymyxin B-treated mouse showed a higher ratio (~20%) of dually labeled survivors, compared with the mice receiving no antibiotics (Fig. 4d, e). By contrast, the other three antibiotics including vancomycin, cefotaxime, and metronidazole all resulted in impaired transplantation efficacies. These data suggest that choosing specific antibiotics for preconditioning may be important during FMT. Other procedures that might improve the subsistence of transplanted bacteria, different microbiota administration methods, for example, could also be evaluated by this method.

### Discussion

We have developed STAMP, a sequential metabolic labeling method for monitoring the survival and metabolic activity of transplanted microbiotas during FMT in vivo. A group of bacterial genera were found to be enriched in the survived transplants, suggesting the effects of FMT might be exerted by a particular subpopulation of the microbiota. This finding may lead to the development of a standardized donor microbiota, which is

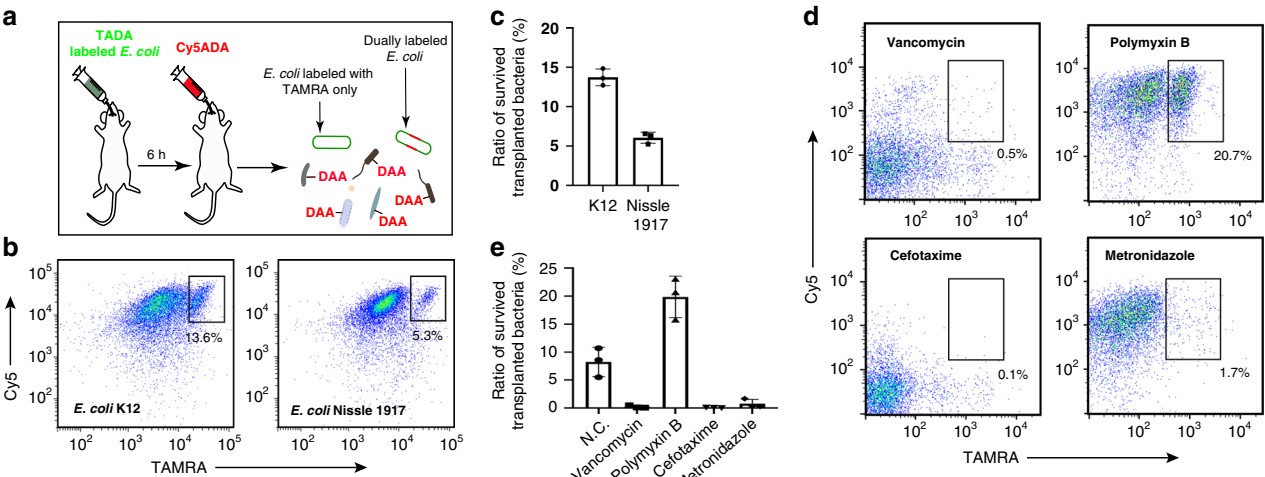

**Fig. 4** Evaluation of the survival of specific bacteria, and the antibiotic preconditioning effects. **a** Schematic showing the experimental procedures for assessing the viability of *E. coli* during transplantation using STAMP. **b** Flow cytometry analysis of the transplanted *E. coli* strains K12 (left) and Nissle 1917 (right). The inserted numbers indicate the survival rates. **c** Statistical analysis of the survival rates of transplanted *E. coli* K12 and Nissle 1917. Mean ± s.d. are presented for n = 3. **d** Flow cytometry analysis of the STAMP-labeled gut microbiota from recipient mice with preconditioning of vancomycin, polymyxin B, cefotaxime, or metronidazole. The inserted numbers indicate the survival rates. **e** Statistical analysis of the survival rates of the transplanted microbiotas from recipient mice with antibiotic preconditioning. Mean ± s.d. are presented for n = 3

composed of specific microbial strains, and more practical to use. We further showed the feasibility of STAMP in studying fecal microbiotas from both mouse and human, which should facilitate the use of STAMP in more clinically relevant settings. Moreover, STAMP allowed us to evaluate the survival of specific bacterial strains in transplantation, and different preconditioning treatments towards the goal of increasing the survival of transplanted bacteria in mouse models. Finally, the division modes of gut microbes were revealed, showing the potential of STAMP for studying basic microbiology in vivo. The translatable knowledge gained from mouse FMT model will help us approach a deeper understanding of this microbial therapy. We believe STAMP is a helpful strategy for studying the gut microbiota, deciphering the complex biology of FMT, and potentially improving its efficacy in treating patients.

## Methods

**Reagents**. The fluorophore NHS esters were purchased from Okeanos Technology (Beijing, China). Vancomycin hydrochloride, metronidazole, polymyxin B sulfate, and cefotaxime sodium were bought from Sangon Biotech (Shanghai, China). And other chemicals, not noted above, were from Sigma-Aldrich (St. Louis, MO, USA).

**Mice and bacterial strain**. Male 6-week-old C57BL/6 SPF mice were obtained from Jie Si Jie Laboratory Animals (Shanghai, China). All mice were bred in the animal facility of Renji Hospital in a temperature-controlled (25 °C) facility with a 12 h light/dark cycle, and received a standard chow diet with free access to clean water. The feed and water were changed every morning to keep fresh. Each group of mice was bred in separate cages. *E. coli* K12 (HfrH) was from China Center of Industrial Culture Collection (Beijing, China). Nissle 1917 was kindly provided by Dr. Jinyao Liu's lab from Shanghai Jiao Tong University School of Medicine, which was originally from German Collection for Microorganisms and Cell Cultures (strain DSM 6601).

**FDAA probes**. FDAA without protecting groups on the carboxyl group were purchased from Chinese Peptide Company (Hangzhou, China). FDAA probes with protecting groups on the carboxyl groups were custom-synthesized by Scilight Biotechnology (Beijing, China).

**Collection of donor mouse gut microbiota labeled with FDAA probes**. The C57BL/6 mice received $2 \times 200\,\mu l$ of 1 mM TADA-amide in PBS by oral gavage with an interval of 3–5 h. The mice were then sacrificed and their gut microbiotas were collected according to a published procedure[17]. Briefly, the mouse intestines (cecum) were dissected with a pair of 4.5-in. iris scissors in 1 ml of degassed phosphate buffer saline (dPBS). The minced tissues and digesta were filtered with cell strainers to remove most of the non-bacterial debris. The bacterial pellets were then washed with $2 \times 1.5$ ml dPBS by centrifugation ($10,000 \times g$, 2 min), and resuspended in dPBS to reach an optical density at 600 nm ($OD_{600}$) of 3.0. To perform propidium iodide (PI) staining, PI was added to the dPBS for cecum dissection to a final concentration of 1 µg/ml, and the gut bacteria were stained for ~10 min during the dissection process. All steps were performed in an anaerobic chamber (Concept 400, Baker Ruskinn, UK).

**FISH labeling**. The FDAA labeled microbiota was washed and resuspended in PBS ($OD_{600} = 1$). An equal volume of EtOH was added into the suspension to fix the bacteria, which was then stored at −20 °C for at least 48 h. The bacteria were spun down and resuspended in a hybridization buffer (0.9 M NaCl, 2 mM Tris (pH 7.5), 0.01% SDS, and 30% formamide). FAM-labeled FISH probe (CFB719, 5′-AGC TGC CTT CGC AAT CGG-3′) was added to the sample with a final concentration of 5 ng/µl and incubated at 46 °C for 4 h. Bacteria were then washed consecutively with two different buffers (0.9 M NaCl, 20 mM Tris (pH 7.5), 0.01% SDS, 30% formamide) and (0.9 M NaCl, 20 mM Tris (pH 7.5), 0.01% SDS), each for 15 min at 48 °C. Cells were resuspended in Tris buffer (20 mM Tris, 25 mM NaCl, pH 7.5) before analysis with fluorescence microscopy.

**FMT and collection of recipient mouse gut microbiota**. The resuspended gut microbiota (200 µl) from the donor mouse labeled with TADA-amide was intragastrically administered to another mouse. Six hours after the gavage, the mouse received $2 \times 200\,\mu l$ of 1 mM Cy5ADA-amide in PBS by oral gavage with an interval of 3 h. The gut bacteria from the cecum of the recipient mouse were collected using the methods described above.

**Mouse fecal microbiota collection and use in FMT**. The fecal pellets from the donor mouse were collected 4–10 h following the first FDAA-gavage, and washed with 1.5 ml dPBS. An aliquot of the resuspended bacteria was used for fluorescence microscopy, and the rest of the bacteria were combined and stored anaerobically

until being used in FMT (200 µl of bacterial suspension used in gavage for each mouse, $OD_{600} = 3.0$).

**Labeling of human fecal microbiota with FDAA**. One gram of freshly collected fecal sample from a healthy volunteer was suspended in 50 ml of dPBS by gentle pipetting and brief vortex (30 s). Large particles were allowed to settle to the bottom of the tubes. The supernatant was then diluted (1:1000) with dPBS, then 100 µl of the diluted suspension was added into the modified Gifu anaerobic liquid medium[17]. The fecal sample was anaerobically incubated at 37 °C for 3 days, and then TADA-amide was added to the medium (300 µM) and cultured for another 2 days. The bacteria were then washed twice and analyzed by fluorescence microscopy.

**Antibiotic treatments**. SPF C57BL/6 mice were randomly assigned into four groups, and each group of mice received treatment with different antibiotics before FMT. Negative control mice received no antibiotic treatments. Each group of mice were respectively given vancomycin (1 mg/ml), metronidazole (1 mg/ml), polymyxin B (1 mg/ml), or cefotaxime (2 mg/ml) in 15 ml daily drinking water for 10 days. The water was changed every day to keep fresh. The day before FMT (36 h at least), antibiotic treatment was stopped by supplementing water without any antibiotics.

***E. coli* labeling with FDAA probes and transplantation**. Two *E. coli* strains, K12 and Nissle 1917, were cultured in LB media at 37 °C till mid-exponential phase, respectively. TADA-amide was then added to the culture media to a final concentration of 300 µM. The bacteria were labeled for 6 h and then washed with PBS twice and resuspended in PBS ($OD_{600} = 2.0$). Transplantation (200 µl suspension of each strain) and a second round of labeling with Cy5ADA-amide were performed as described above in FMT.

**Flow cytometry**. Flow cytometry analyses and sorting of the FDAA probe labeled microbiota samples were performed on CytoFLEX (Beckman Coulter Life Sciences, Indianapolis, IN, USA) and Aria II flow cytometer (BD Biosciences, San Jose, CA, USA). FlowJo software (V 10.0.8r1) was used for data analyses. Labeled bacteria were identified with flow cytometry plots of logFSC versus logSSC and then gated on fluorescence. For each sample, 15,000 events were collected for analysis (debris and doublets excluded). For cell sorting, $4.0 \times 10^6$ of the double-labeled bacteria were collected and used in 16S rDNA sequencing.

**Fluorescence microscopy**. Labeled bacteria were inoculated onto agarose pads (1.5% w/v in PBS, ~1 mm in thickness) and covered with glass coverslips. Confocal microscopy was performed on a Nikon A1R laser scanning confocal microscope. Samples were excited with 555 nm for TAMRA, and 639 nm for Cy5, and the emission was detected using corresponding emission filters.

**Frozen-sectioning and imaging of the administered microbiota**. The gut microbiota collected from the cecum of a TADA-amide-gavaged mouse was intragastrically administered to another mouse. Six hours after the gavage, the mouse was sacrificed, and the intestines were dissected and fixed in 4% (w/v) paraformaldehyde in PBS for 4 h. Following dehydration in PBS containing 30% (w/v) sucrose overnight at 4 °C, the intestines were mounted in tissue freezing medium, frozen at −80 °C and sectioned (10 µm in thickness). TAMRA-labeled transplanted bacteria were directly visualized following counterstaining with Hoechst 33342 (1 µg/ml) on a Nikon A1R laser scanning confocal microscope.

**DNA extraction and PCR amplification**. DNA from the recipient mouse cecum's bacterial samples before sorting, and the bacteria sorted by FACS were extracted using an Omega Bacterial DNA Kit (Omega Bio-tek, Norcross, GA, USA) according to manufacturer's protocol. The V3–V4 region of the bacteria 16S rDNA was amplified by PCR (95 °C for 3 min, followed by 27 cycles at 95 °C for 30 s, 53 °C for 30 s, and 72 °C for 45 s and a final extension at 72 °C for 10 min) using primers 338F 5′-ACTCCTACGGGAGGCAGCAG-3′ and 806R 5′-GGACTACHVGGGTWTC-TAAT-3′ (Supplementary Table 1). PCR reactions were performed in triplicate 20 µl mixture containing 2 µl of 10× PCR buffer, 2.5 ml of 2.5 mM dNTPs, 0.8 µl of each primer (5 µM), 0.4 µl of rTaq polymerase, and 10 ng of template DNA.

**16S rDNA sequencing**. Amplicons were extracted from 2% agarose gels, purified with an AxyPrep DNA Gel Extraction Kit (Axygen Biosciences, Union City, CA, USA) according to the manufacturers' protocol, and quantitated using QuantiFluo-ST (Promega, Madison, WI, USA). Purified amplicons were then pooled in equimolar and paired-end sequenced ($2 \times 250$) on an Illumina MiSeq platform (Illumina, San Diego, CA, USA) following the standard protocol.

**Processing of sequencing data**. Raw fastq files were de-multiplexed, quality-filtered using QIIME (version 1.17) with the following criteria: (1) The 300 bp reads were truncated at any site with an average quality score <20 over a 50 bp sliding window and discarded the truncated reads <50 bp. (2) Exact barcode matching, two nucleotide mismatches in primer matching, or reads containing ambiguous characters were removed. (3) Only sequences that overlap >10 bp were assembled based on their

overlapping sequence; reads not assemblable were discarded. Operational taxonomic units (OTUs) were then clustered with 97% similarity cutoff (UPARSE, version 7.1), chimeric sequences were identified and removed using UCHIME. The taxonomy of each 16S rRNA gene sequence was analyzed by RDP Classifier (http://rdp.cme.msu.edu/) against the SILVA (SSU115) 16S rDNA database with a confidence threshold of 70%.

**Compliance with ethical standards**. The fecal microbiota donor had given informed consent for the experiment, and the use of human microbiota samples was approved by the Ethics Committee at Renji Hospital of Shanghai Jiao Tong University School of Medicine. All animal experiments were carried out in accordance with guidelines of ethical regulations for animal testing and research, approved by the Institutional Animal Care and Use Committee of the Shanghai Jiao Tong University School of Medicine.

**Reporting summary**. Further information on experimental design is available in the Nature Research Reporting Summary linked to this article.

## Data availability
Sequencing data of the cecal microbiota before and after FACS have been deposited in the Sequence Read Archive with BioSample accessions SAMN10907938 and SAMN10907939. The source data underlying Figs. 2c, 3c, 4c, e and Supplementary Figs. 4 and 5 are provided as a Source Data file. Other relevant data are available from the corresponding authors on request.

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

## Acknowledgements
The authors are grateful to the National Science Foundation of China (21807070, 21735004, 21672013, 21775128, 21705024, 21521004) for the financial support. The authors thank the Center of Excellence (COE), BD China, Shanghai for their support of cell sorting, and the animal facility at Renji Hospital for their support with the mouse study.

## Author contributions
W.W., X.C., and C.Y. designed the study and prepared the manuscript. W.W., L.L., Y.D., and X.P. performed and analyzed experiments. Y.S. contributed intellectually to the analysis and interpretation of the data. W.W., X.C., and C.Y. wrote the manuscript.

## Additional information

**Competing interests:** The authors declare no competing interests.

