## [Peer Review File · Nature Communications]

Reviewers' comments:

Reviewer #1 (Remarks to the Author):

In this manuscript the authors elegantly utilize fluorescent D-amino acids (FDAAs) in order to probe, distinguish AND enrich the survival and the growth of individual species within the host following a fecal microbiota transplantation (FMT) procedure. As the authors also noted, other genetic labeling approaches are limited to study gut microbiota because of their generally difficult-to-culture status and their general strict dependency to anaerobic conditions (genetically encoded fluorescent proteins do not mature absent molecular oxygen).

Furthermore, I am excited to see that fluorescent D-amino acid technology being applied to such an important and interesting medical problem. In general, I thought the manuscript was concise, clear and well written.

In these ways, I find this manuscript novel and of interest to others in the community and the wider field of microbiology and medicine. This is despite some experimental concerns (expressed below), which I can excuse (as long as it is discussed appropriately in the main text) because I see this manuscript as a pioneering study opening the way to apply STAMP in addressing many other similar microbiome related questions. Therefore, I believe this manuscript will influence thinking in the microbiota field and will facilitate the use of sequential D-amino acid-based metabolic probe labeling (STAMP) to rapidly study the health of microbiota in applications that will exceed FMT procedures.

This said I have couple of concerns that I would like to see being addressed.

Major point:

***The experiments leading to Figure 2C and the interpretation "The unlabeled population was very likely dead bacteria, since it required metabolic activity to get the probe incorporated. "

This observation is more likely because TAMRA is a very large fluorophore; too large to get across the outer membranes (a semi-permeable barrier that is supposed to limit molecules larger than 500 Daltons) of some of the gram-negative bacterial species. I would expect this being worse with a larger Cy5- containing FDAAs. I would like to see the coverage specifically focused on outer-membrane containing gram-negative bacteria.

Specifically, could the authors quantify the coverage in Fig S3 like they did Figure 2C? and comment on if gram-negative bacteria are enriched at the fraction of cells that are not being labeled by Cy5- containing FDAAs or DAA-TAMRA (also known as TADA, see below)?

If there is indeed an enrichment of outer-membrane containing bacteria that do not get labeled by these FDAAs, this is not a deal-breaker as long as the authors mention this possibility in the manuscript.

They could also suggest using smaller FDAAs in the future. Reference 20 (Hudak et al) uses HADA because it is a small FDAA labeling both Gram-negative and Gram-positive bacteria. If the authors needed FDAAs brighter and/or red-shifted than HADA because of any detection reasons, I would also recommend them to refer to new set of FDAAs that are bright and red-shifted yet small to label both Gram-negative & Gram-positive bacteria.

(<https://www.ncbi.nlm.nih.gov/pmc/articles/PMC5628581/>)

This main point is especially important when drawing conclusions like the following:

"In contrast to these survived transplants, much of the bacterial population was single-colored. Those showing only TAMRA signal were probably the transplanted bacteria that died during FMT, since they had little metabolic activities (Cy5 signals) in the recipient's gut, and the bacteria labeled with Cy5 alone were most likely the recipient's original gut microbiota (Fig. S5)."

This is a valuable and interesting study; however for people wanting to adopt STAMP in the future

I think it is essential to accompany conclusions like this with the aforementioned caveats, that is the choice of probes could bias labeling some set of species over others.

Minor points:

***Though I like the term “sequential D-amino acid-based metabolic probe labeling” (STAMP); it is renaming a concept that was first described as “virtual-time lapse microscopy” in (<https://doi.org/10.1002/ange.201206749>) and then further utilized in (<https://www.nature.com/articles/s41564-017-0029-y>).

When a method that already exists is being renamed; the authors should also acknowledge the original method. In other words, instead of:

“Here, we report a sequential D-amino acid-based metabolic probe labeling (STAMP) strategy,”
Something like:

“Inspired by previous approaches (references) here, we report a sequential D-amino acid-based metabolic probe labeling (STAMP) strategy...”

***Similarly, some of the molecules the authors have shown are already published and should be cited appropriately. Specifically:

1-All the molecules the authors made should be referred as Fluorescence D-amino acids or FDAAs as coined in (<https://doi.org/10.1002/ange.201206749>)

2- DAA-TAMRA is an FDAA that is made before and referred as TAMRA-amino-d-alanine or simply TADA (<https://www.nature.com/articles/s41564-017-0029-y>).

The authors could adopt this naming as TADA, TADA-amide and TADA-ester. Alternatively, they could just refer the name TADA and the paper; when the molecules are first introduced.

3- Amidated FDAAs were first discussed here (<https://pubs.acs.org/doi/abs/10.1021/ja505668f>)

***I noticed 1-2 grammatical errors in the manuscript that I trust could be fixed at the editorial level.

Reviewer #2 (Remarks to the Author):

Summary

The authors present a compelling application of cutting-edge labeling technologies to the examination of fecal microbiota transplants (FMTs) in a mouse model. D-amino acids clearly represent a stable, potent, and tractable system for tracking gut commensal bacteria in vivo such that FMTs can be regularly and easily evaluated for maximum engraftment and viability in recipients.

Major concerns:

While several parameters in the STAMP protocol have been optimized herein, there are quite a few other easily attainable experimental strategies that would boost the appeal of this methodology to the human FMT research community:

1. The authors have only examined specific pathogen-free mouse microbiota, and have failed to test DAA staining of human donor stool. This is a completely safe and noninvasive type of sample to attain, and is the gold standard when considering FMT. Demonstrating the feasibility of DAA staining on human fecal microbes is a realistic expectation for a manuscript of this caliber.

2. Furthermore, all of the presented mouse experiments use cecal contents as inocula rather than homogenized fecal pellets. Does the STAMP protocol work just as effectively with fecal samples as with intestinal digesta harvested from euthanized animals? If not feasible with fecal samples, then that would render this a completely ineffective method for the FMT field.

Minor concerns:

3. The majority of data presented are from a 12hr post-inoculation time point. While the authors write that this gives "transplanted bacteria enough time for 'adopting' the new environment", they offer no data to support this claim. A time course would help determine both (A) the rate at which original DAA-TAMRA stain decays and (B) the optimal time point at which donor bacteria are most robust in replicating and stabilizing in the recipient gut.

Reviewer 1:**Comments:**

In this manuscript the authors elegantly utilize fluorescent D-amino acids (FDAAs) in order to probe, distinguish AND enrich the survival and the growth of individual species within the host following a fecal microbiota transplantation (FMT) procedure. As the authors also noted, other genetic labeling approaches are limited to study gut microbiota because of their generally difficult-to-culture status and their general strict dependency to anaerobic conditions (genetically encoded fluorescent proteins do not mature absent molecular oxygen).

Furthermore, I am excited to see that fluorescent D-amino acid technology being applied to such an important and interesting medical problem. In general, I thought the manuscript was concise, clear and well written.

In these ways, I find this manuscript novel and of interest to others in the community and the wider field of microbiology and medicine. This is despite some experimental concerns (expressed below), which I can excuse (as long as it is discussed appropriately in the main text) because I see this manuscript as a pioneering study opening the way to apply STAMP in addressing many other similar microbiome related questions. Therefore, I believe this manuscript will influence thinking in the microbiota field and will facilitate the use of sequential D-amino acid-based metabolic probe labeling (STAMP) to rapidly study the health of microbiota in applications that will exceed FMT procedures.

This said I have couple of concerns that I would like to see being addressed.

Major point:

The experiments leading to Figure 2C and the interpretation “The unlabeled population was very likely dead bacteria, since it required metabolic activity to get the probe incorporated. “

This observation is more likely because TAMRA is a very large fluorophore; too large to get across the outer membranes (a semi-permeable barrier that is supposed to limit molecules larger than 500 Daltons) of some of the gram-negative bacterial species. I would expect this being worse with a larger Cy5- containing FDAAs. I would like to see the coverage specifically focused on outer-membrane containing gram-negative bacteria.

Specifically, could the authors quantify the coverage in Fig S3 like they did Figure 2C? and comment on if gram-negative bacteria are enriched at the fraction of cells that are not being labeled by Cy5- containing FDAAs or DAA-TAMRA (also known as TADA, see below)?

If there is indeed an enrichment of outer-membrane containing bacteria that do not get labeled by these FDAAs, this is not a deal-breaker as long as the authors mention this possibility in the manuscript.

They could also suggest using smaller FDAAs in the future. Reference 20 (Hudak et al) uses HADA because it is a small FDAA labeling both Gram-negative and Gram-positive bacteria. If the authors needed FDAAs brighter and/or red-shifted than HADA because of any detection reasons, I would also recommend them to refer to new set of FDAAs that are bright and red-shifted yet small to label both Gram-negative & Gram-positive bacteria. (<https://www.ncbi.nlm.nih.gov/pmc/articles/PMC5628581/>)

This main point is especially important when drawing conclusions like the following: “In contrast to these survived transplants, much of the bacterial population was single-colored. Those showing only TAMRA signal were probably the transplanted bacteria that died during FMT, since they had little metabolic activities (Cy5 signals) in the recipient’s gut, and the bacteria labeled with Cy5 alone were most likely the recipient’s original gut microbiota (Fig. S5).”

This is a valuable and interesting study; however for people wanting to adopt STAMP in the future I think it is essential to accompany conclusions like this with the aforementioned caveats, that is the choice of probes

could bias labeling some set of species over others.

Author Response:

We greatly appreciate the comments and suggestions from the reviewer. To investigate whether Gram-negative bacteria were enriched in the unlabeled population, we used a *Bacteroidetes*-targeting (the major Gram-negative phylum in gut microbiota) FISH probe to stain the FDAA-labeled microbiota. We found that, besides the dead bacteria that were unlabeled by FDAA (showed by staining with propidium iodide, Fig. S2), there were indeed some *Bacteroidetes* bacteria that are unlabeled by FDAA (Fig. S3). Even though we did not find any significant difference of the labeling coverage between TADA-amide and Cy5ADA-amide labeled microbiota (Fig S5b), as suggested by the reviewer, a FDAA containing a smaller fluorophore might improve the labeling of this small group of Gram-negative bacteria. However, considering the low ratio of this unlabeled living bacteria and the fact that most FDAAs with relatively small fluorophores are green-emitting, where the gut bacteria have strong autofluorescence, we decided to use TADA and Cy5ADA in the following studies. We have discussed these points in the revised manuscript (Page 2, paragraph 1).

Minor points:

1. *Though I like the term “sequential D-amino acid-based metabolic probe labeling” (STAMP); it is renaming a concept that was first described as “virtual-time lapse microscopy” in (<https://doi.org/10.1002/ange.201206749>) and then further utilized in (<https://www.nature.com/articles/s41564-017-0029-y>).*

When a method that already exists is being renamed; the authors should also acknowledge the original method. In other words, instead of:

“Here, we report a sequential D-amino acid-based metabolic probe labeling (STAMP) strategy,”

Something like:

“Inspired by previous approaches (references) here, we report a sequential D-amino acid-based metabolic probe labeling (STAMP) strategy...”

Author Response:

We thank the reviewer for the comments. We have modified the expression in the manuscript to “Inspired by a chronological FADA labeling strategy reported previously (Ref), here we report a sequential D-amino acid-based metabolic probe labeling (STAMP) strategy” (Page 1, paragraph 3)

2. *Similarly, some of the molecules the authors have shown are already published and should be cited appropriately. Specifically:*

1-All the molecules the authors made should be referred as Fluorescence D-amino acids or FDAAs as coined in (<https://doi.org/10.1002/ange.201206749>)

2- DAA-TAMRA is an FDAA that is made before and referred as TAMRA-amino-d-alanine or simply TADA (<https://www.nature.com/articles/s41564-017-0029-y>).

The authors could adopt this naming as TADA, TADA-amide and TADA-ester. Alternatively, they could just refer the name TADA and the paper; when the molecules are first introduced.

3- Amidated FDAAs were first discussed here (<https://pubs.acs.org/doi/abs/10.1021/ja505668f>)

I noticed 1-2 grammatical errors in the manuscript that I trust could be fixed at the editorial level.

Author Response:

We thank the reviewer for the comments. We have changed the naming of the probes as suggested by the reviewer throughout the manuscript, and added the new references mentioned above. The grammatical errors have been also corrected.

Reviewer 2:**Comments:**

The authors present a compelling application of cutting-edge labeling technologies to the examination of fecal microbiota transplants (FMTs) in a mouse model. D-amino acids clearly represent a stable, potent, and tractable system for tracking gut commensal bacteria in vivo such that FMTs can be regularly and easily evaluated for maximum engraftment and viability in recipients.

Major concerns:

While several parameters in the STAMP protocol have been optimized herein, there are quite a few other easily attainable experimental strategies that would boost the appeal of this methodology to the human FMT research community:

1. The authors have only examined specific pathogen-free mouse microbiota, and have failed to test DAA staining of human donor stool. This is a completely safe and noninvasive type of sample to attain, and is the gold standard when considering FMT. Demonstrating the feasibility of DAA staining on human fecal microbes is a realistic expectation for a manuscript of this caliber.

Author Response:

We thank the reviewer for the suggestion. To demonstrate the feasibility of DAA labeling on human fecal microbes, we used a previously reported *in vitro* human gut microbiota culture system to cultivate and label the human fecal bacteria with FDAA. Following 3 days of anaerobic culture and then 2 days of FDAA labeling, the bacteria in the human feces were strongly labeled by the FDAA probe (Fig. S11). This demonstrates the feasibility of DAA labeling with human microbiotas.

2. Furthermore, all of the presented mouse experiments use cecal contents as inocula rather than homogenized fecal pellets. Does the STAMP protocol work just as effectively with fecal samples as with intestinal digesta harvested from euthanized animals? If not feasible with fecal samples, then that would render this a completely ineffective method for the FMT field.

Author Response:

We thank the reviewer for the comments. To evaluate the feasibility of STAMP with fecal bacteria, we collected the fecal pellets of a donor mouse from 4 h to 10 h after the DAA-gavage. After a quick washing with degassed PBS, the bacteria were analyzed by fluorescence microscopy, and they were found to be highly labeled by FDAA probes (Fig S10a). The combined fecal bacteria (kept in an anaerobic environment during the collection) were used in FMT as donor microbiota. The second round of FDAA labeling, using the same

protocol as used for cecal microbiota, revealed a subgroup of gut bacteria that were doubly labeled (Fig S10b), demonstrating the feasibility of our STAMP method for tracking the viability of fecal bacteria in FMT.

Minor concerns:

3. The majority of data presented are from a 12hr post-inoculation time point. While the authors write that this gives “transplanted bacteria enough time for ‘adopting’ the new environment”, they offer no data to support this claim. A time course would help determine both (A) the rate at which original DAA-TAMRA stain decays and (B) the optimal time point at which donor bacteria are most robust in replicating and stabilizing in the recipient gut.

Author Response:

We thank the reviewer for the comments. As suggested by the reviewer, a time course of the transplantation was investigated. Following transplanting the donor microbiota, different waiting times (3 h, 6 h, 9 h) were used before performing the second round of FDAA labeling (requires another 6 h). The following two-color imaging showed that the donor microbiota's fluorescence was similar in the 3 h and 6 h groups, and decreased quickly between 6-9 h (Fig. S6). To give the transplants more time for adopting their new host, we therefore chose the 6 h waiting time between the FMT and the second round of FDAA labeling.

The optimal time point at which donor bacteria are most robust in replicating and stabilizing in the recipient gut was very difficult to determine by our method because, while the second labeling became stronger during the second FDAA labeling, the donor fluorescence decays gradually as time goes on. And different bacteria have quite different generation times, therefore, determining the optimal time points for donating bacterial replication is impractical using our method.

REVIEWERS' COMMENTS:

Reviewer #1 (Remarks to the Author):

I am satisfied with the changes the authors made

Reviewer #2 (Remarks to the Author):

The FDAA-based STAMP method presented herein is a novel application of an underutilized technique for examining transferred gut commensal microbes *in vivo*. As the authors have demonstrated that this technique is not only feasible with endogenous mouse cecal bacteria, but also human fecal bacteria, we believe that this manuscript is now suitable for publication in NComm.

REVIEWERS' COMMENTS:

Reviewer #1 (Remarks to the Author):

I am satisfied with the changes the authors made

Our response: We thank the positive feedback from the reviewer.

Reviewer #2 (Remarks to the Author):

The FDAA-based STAMP method presented herein is a novel application of an underutilized technique for examining transferred gut commensal microbes in vivo. As the authors have demonstrated that this technique is not only feasible with endogenous mouse cecal bacteria, but also human fecal bacteria, we believe that this manuscript is now suitable for publication in NComm.

Our response: We thank the reviewers' valuable time reviewing our manuscript, and we are pleased that the revision of the work is well appreciated.